# ADP-Ribosylation as Post-Translational Modification of Proteins: Use of Inhibitors in Cancer Control

**DOI:** 10.3390/ijms221910829

**Published:** 2021-10-07

**Authors:** Palmiro Poltronieri, Masanao Miwa, Mitsuko Masutani

**Affiliations:** 1Institute of Sciences of Food Productions, National Research Council of Italy, CNR-ISPA, Via Monteroni, 73100 Lecce, Italy; 2Nagahama Institute of Bio-Science and Technology, Nagahama 526-0829, Japan; m_miwa@nagahama-i-bio.ac.jp; 3Department of Molecular and Genomic Biomedicine, CBMM, Nagasaki University Graduate School of Biomedical Sciences, Nagasaki 852-8523, Japan

**Keywords:** ADP-ribosyl transferase (ART), poly ADP-ribose polymerase (PARP), ADP-ribose (ADPR), sirtuin (SIRT), poly ADP-ribose glycohydrolase (PARG), ADP-ribose hydrolase (ARH), macro-domain (MACRO)

## Abstract

Among the post-translational modifications of proteins, ADP-ribosylation has been studied for over fifty years, and a large set of functions, including DNA repair, transcription, and cell signaling, have been assigned to this post-translational modification (PTM). This review presents an update on the function of a large set of enzyme writers, the readers that are recruited by the modified targets, and the erasers that reverse the modification to the original amino acid residue, removing the covalent bonds formed. In particular, the review provides details on the involvement of the enzymes performing monoADP-ribosylation/polyADP-ribosylation (MAR/PAR) cycling in cancers. Of note, there is potential for the application of the inhibitors developed for cancer also in the therapy of non-oncological diseases such as the protection against oxidative stress, the suppression of inflammatory responses, and the treatment of neurodegenerative diseases. This field of studies is not concluded, since novel enzymes are being discovered at a rapid pace.

## 1. Introduction

Post-translational modifications (PTM) of proteins require the activity of enzyme writers and erasers, and their presence is recognized by readers, protein modules, or structures that bind to specific chemical structures and organize protein–protein interactions. In PTM reactions, the amino acids that are modified may be available or may already be involved in other modifications. Therefore, different types of PTM may interfere with the same amino acids.

The purpose of this review is to update the current knowledge on ADP-ribosylation reactions. There are several types of enzymes, broadly defined as polyADP-ribosylating (PARylating) and monoADP-ribosylating (MARylating) enzymes. In the following paragraph we introduce the polymer-forming writers, the poly(ADP-ribose) polymerases. Then, a short presentation introduces the diphtheria-toxin-like ADP-ribose transferases (ARTDs), the cholera-toxin-like ARTCs, and the MARylating sirtuins. Several ADP-ribosylating enzymes are linked to cancer development, and the review discusses the proposed applications of specific ART inhibitors for the containment of cancers and their therapeutic potential.

## 2. ADP-Ribosylation

Two main groups of enzyme writers of protein ADP-ribosylation are described. MARylating enzymes attach one unit of ADPR, while PARylating enzymes synthesize and attach long PAR polymers. Scaffolding functions are ascribed to certain mono(ADP-ribose) (MAR) and poly(ADP-ribose) (PAR) structures. Several types of proteins may read these domains and are recruited to MAR and PAR sites. Enzymes that erase the modification in part or completely are described in detail. The ADP-ribosylation of phosphorylated ends of RNA and DNA will not be discussed in the review.

### 2.1. Enzymes Involved in ADP-Ribosylation of Proteins: The Writers

The writers use NAD^+^ as a substrate to transfer the ADP-ribose (ADPR) moiety to target amino acids and release nicotinamide (NAM): when one unit of ADPR is attached, they are named mono-ADP-ribosyl-transferases (mARTs) and the reaction is called MARylation [1,2] (see Figure 1); when a polymer of various units of ADPR is formed, linear or branched, they are named poly(ADP-ribose) polymerases (PARPs), and this modification of proteins is called PARylation [1,2] (Figure 2).

ADP-ribosylation reactions are present in all forms of life, such as invertebrates, fungi, bacteria, and viruses [3,4]. The main group of ART genes coding for MAR and PAR writers is clustered into seven clades [4]. Many bacterial pathogens possess enzymes performing this modification to deregulate cellular functions [5]. For instance, the cholera toxin MARylates arginine on Gs alpha subunit, to render GTPase’s activity constitutive. The diphtheria toxin MARylates diphthamide (a modified histidine) on elongation factor 2 (EF-2) to inhibit protein synthesis. In humans, there are seventeen transcribed proteins belonging to a diphtheria-toxin-like group, named ADP-ribosyl transferases D-type (ARTDs), involved in MAR/PAR modifications [6]. The biological functions of PARylation have been extensively reviewed [7,8]. PARP1, PARP2, and tankyrases (PARP5a/ARTD5, PARP5b/ARTD6, TNKS) perform PARylation, and PARP1 and PARP2, in particular, can synthesize long, branched polymers. PARP3 and Vault-PARP4 attach mainly one unit of ADPR. The PARylating PARPs are recognized by the histidine-tyrosine-glutamate (H-Y-E) triad, in which glutamate has a catalytic role in PAR synthesis [6]. PARP1 and PARP2 can heterodimerize: the DNA-binding domain and BRCA1 C-terminal (BRCT) domain of PARP1 are responsible for this interaction [8]. The dimer partners ADP-ribosylate each other, which may be relevant for the efficient DNA repair of single-strand and double-strand breaks [7,8]. PARP3 binds to PARP1 and activates PARP1 by MARylation even in the absence of DNA damage [9,10,11]. 

The nomenclature of enzymes has changed over time [12], so that certain enzymes, when named as ARTD or as PARP, maintained the same number, while for others, when using the name PARP or the name ARTD, the number has changed. Therefore, when discussing the enzymes in which the numbering is different, this review will use both names to avoid confusion.

As for the human MARylating ARTs, their members are characterized by the presence in the triad motif of isoleucine (I), leucine (L), threonine (T), valine (V), or tyrosine (Y) instead of glutamate [6]; all these ARTs have enzymatic activity [1,13] except ARTD13, in which the catalytic domain is mutated, but has developed new antiviral properties. As for ARTD9, considered to be inactive, it forms a complex with Deltex (DTX3L) protein, a ubiquitylating enzyme, and the complex MARylates glycine76 on ubiquitin: this reaction is called PARylation-dependent ubiquitylation (PARdU) [1,14]. 

Among cholera-toxin-like mono(ADP-ribosyl) transferases (ARTCs), four genes are expressed in humans, but only ARTC1 and ARTC5 are active [15].

In addition to these PARP-domain-containing enzymes, there are few sirtuins with MARylating activity. SIRT4, SIRT6, and SIRT7 perform a specific MARylation of selected targets and in this way regulate cell metabolism and transcription [15]. Sirtuins are members of histone deacetylases. Sirtuins use NAD^+^ to remove acyl and acetyl groups from lysine, forming 2′-O-acyl-ADP-ribose [1,16]. Among MARylating sirtuins, SIRT4 MARylates glutamate dehydrogenase (GDH) [17] in mitochondria, while SIRT6 and SIRT7 are localized in the nuclei and MARylate a large set of target proteins. 

### 2.2. Amino Acids Modified by MAR/PAR

In the pioneering years, PARP1 was principally thought to be involved in modification of acidic amino acids, with glutamate and aspartate as acceptors for auto-modification. Recently, it was shown that serine residues of target proteins such as histones are the most abundant acceptor sites for PARylation, especially after DNA damage. This is made possible by the interaction of PARP1 with histone PARylation factor 1 (HPF-1), that changes the tertiary structure of PARP1 and PARP2, granting them specificity for serines. In the absence of HPF1, PARP1 preferentially auto-modifies itself, becoming unavailable for HPF1 interaction [18,19]. During the DNA damage repair, PARP1 and PARP2 modify serine residues [1], but also, to a lesser extent, tyrosine, lysine, and acidic residues [20]. Amino acid targets that can be modified include lysine (by PARP1, PARP3, ARTD10, ARTD11, and ARTD15/PARP16), aspartate and glutamate (by PARP1/2/3, ARTD8/PARP14, ARTD10, ARTD11, ARTD12, ARTD15/PARP16, ARTD17/PARP6), serine (PARP1/2), and cysteine (by ARTD11, ARTD12, ARTD17/PARP6); additional amino acids that can be modified are arginine and histidine [21,22,23,24]. However, the contribution of enzymatic ADP-ribosylation versus non-enzymatic conjugation reactions requires further studies [25,26]. Concerning the sirtuin-dependent MARylation of amino acids, they transfer ADP-ribose to arginine, serine, threonine, and cysteine residues of proteins.

## 3. Readers

There are several motifs for readers to recognize in MARylation and PARylation. The PAR-binding motif (PBM) is present in proteins involved in the DNA damage response (DDR), chromatin remodeling, or RNA processing. The PAR-binding zinc finger (PBZ) motif is present in proteins involved in DDR, such as the aprataxin PNK-like factor (APLF), and in the regulation of DNA damage-activated checkpoints, i.e., checkpoints with FHA and RING domains (CHFR). The WWE domain is present in two groups of proteins: E3 ubiquitin ligases such as RNF146 and, in a few cases, ARTDs (PARP1, tankyrases, ARTD8/PARP14, ARTD11). The RNF146/Iduna ubiquitin ligase associates with tankyrase, involved in PARylation-dependent ubiquitylation (PARdU) reactions: tankyrases PARylate their substrate proteins, and then RNF146 recognizes the PAR portion of PARylated proteins by its WWE domain and is allosterically activated to ubiquitylate them for proteasomal degradation [27,28,29,30]. Several tankyrase binding motifs (TBM) act in concert to allow the interaction between RNF146 and the ankyrin repeat clusters (ARC) of tankyrases [31]. An Oligonucleotide/oligosaccharide-binding (OB) fold is present in single strand-DNA binding protein 1 (hSSB1), which relocates to sites of DNA damage by the recognition of PAR polymers, promoting the DDR [32]. The exonuclease EXO1 contains a domain named “protein incorporated later into tight junctions (PilT) N terminus” (PIN). The PIN domain that binds to PAR facilitates EXO1 recruitment to DNA and supports the repair of double-strand breaks of DNA [33]. Other domains may contain the RNA recognition motif (RRM), the serine arginine repeats, and the lysine arginine repeats (SR/KR) motifs, or the Forkhead-associated (FHA) domain (binding to iso-ADP-ribose) and the BRCA1 C-Terminal (BRCT) in the Breast Cancer 1 (BRCA1) antioncogene, binding to (ADPR)_n_ units [32,34]. ADPR-binding macrodomains can behave as readers, as in the case of AF1521 from *Archeglobus fulgidus*, that has been applied to proteomic studies for the enrichment of ADP-ribosylated proteins [35,36,37]. A PAR-recognizing macrodomain is present in a histone H2A variant named H2A1.1 (mH2A1.1). This macrodomain interacts with PAR and may support the repair of DNA by stabilizing the chromatin. MacroH2A1.1 is recruited to DNA regions rich in PAR, stabilizing their negative charges and slowing down PAR degradation, NAD^+^ consumption, and cell death, allowing for an efficient DNA repair after DNA damage [38]. MARylating sirtuins such as SIRT7 attract the ADP-ribose-binding macrodomain of histone H2A1.1 and promote H2A1.1 accumulation in loci containing metabolic genes [39]. ALC1/CHD1L is a macrodomain protein, a PAR-binding Snf2-like ATPase that has oncogenic activity and is frequently overexpressed in hepatocellular carcinoma (HCC) and other types of cancers. ALC1 is also a chromatin remodeler, and a helicase that supports nucleosomal sliding. In homologous recombination-deficient cells such as in BRCA-mutated phenotypes, the loss of ALC1 sensitizes cells to the PARP inhibitor, in spite of various resistance mechanisms, and reduces the chromatin accessibility to DNA repair proteins [32,40]. Since there are several motifs recognizing specific structures, some of them may compete with other PAR binding proteins, inhibiting their interactions; steric inhibition is another mechanism that may occur to impede the PAR binding, making the region inaccessible to the interacting readers. Other macrodomain proteins behave both as readers and as erasers. 

## 4. Erasers

PAR glycohydrolase (PARG) cleaves ADPR/ADPR bonds. PARG is a macrodomain-containing enzyme that efficiently cleaves PAR chains through exo- and endo-glycosyl hydrolytic activity. However, PARG cannot remove the terminal ADPR linked to protein substrates. PARG has an important role in SSB and DSB DNA repair, which was shown using a cell-permeable PARG inhibitor [41] and functional approaches [42]. The deletion or inhibition of eraser enzymes such as PARG may be used to potentiate the synthetic lethality drugs in cancers with mutant phenotypes., and for PARP inhibitor-resistant cancers [43,44,45] or to sensitize cancer chemo- and radiotherapy [46]. PARG interacts with the synaptonemal complex proteins, REC-8/Rec8, and the MRN/X complex [47]. Several macrodomain proteins possess hydrolase activity to cleave the ADPR bound to proteins [48]. The PAR cleaving enzymes include the ADPR hydrolase 3 (ARH3), which can cleave ADPR–ADPR bonds through exoglycosidic and endoglycosidic activities. ARH3 can cleave the ADPR from serine residues [48,49]. MacroD1, MacroD2, and terminal ADP-ribose protein glycohydrolase (TARG) specifically revert MARylation by hydrolyzing the bond between the terminal ADPR bound to acidic residues (Glu, Asp) [50,51]. 

TARG1 performs hydrolase activity toward MAR/PAR as well as *O*-acetyl ADPR (OAADPr) deacetylase activity. Ectonucleotide pyrophosphatase/phosphodiesterase 1 (ENPP1) transforms the last ADPR group, into phosphoribose bound to the proteins leaving 5′AMP. Similarly, nucleoside diphosphates linked to some moiety X (NUDX/NUDT) hydrolases cleave ADP-ribose to produce 5′ AMP and ribose-5-phosphate (R5P) [16,52,53]. Various NUDIX/NUDT enzymes are overexpressed in cancers. In particular, NUDT2 and NUDT5 are linked to aggressive adenocarcinomas and homologous recombination (HR)-positive breast cancers, respectively [52], linking them to ADPR catabolism and the NAD^+^ salvage pathway [54,55]. PAR- and MAR-processing enzymes are often deregulated in cancers. MacroD1 is also named leukemia-related protein 16 (LRP16). An aberrant MacroD1 expression has been associated with leukemia and breast, colorectal, gastric, lung, and liver cancers [53,54]. When overexpressed, MacroD2 is associated with estrogen-independent growth and tamoxifen resistance in breast cancer [50,56]. The overexpression of eraser proteins may disrupt the physiology of the signaling role of ADP-ribosylation reactions: following the activity of MARylating enzymes, a pathological accumulation of erasers will reverse the modification rapidly and subvert the requirement of this PTM in a determined timing and cellular localization, preventing the downstream signaling from being transduced.

## 5. Various Types of Post-Translational Modification of PARP1 and Competition on Amino Acid Substrates

Concerning PARP1, several post-translational modifications regulate its activity. A SUMOylation on K486 renders this lysine unavailable for acetylation, determining a competition between SUMOylation and acetylation modifications of PARP1 [57]. Sirtuin 6 (SIRT6) directly associates with PARP1 and MARylates PARP1 at lysine 521, stimulating PARylation activity and DNA repair [58]. A similar role is ascribed to PARP3, that MARylates PARP1 on a different amino acid. In PARP1, lysine 508 (K508) can be either acetylated by p300/CBP acetylase or methylated by SET7/9. Thus, when K508 is methylated, PARP1 cannot be recruited to PARylate the p50 subunit of transcription factor NF-kB during the inflammatory response. K508 methylation by SET7/9 can be blocked by PARP1 auto-modification, probably due to steric hindrance and charge effects, making this PTM alternative to methylation. Histone methylase SMYD2 modifies lysine 528 on PARP1, enhancing the PARylation activity in response to oxidative stress [59,60]. Histones are displaced from nucleosomes and chromatin by PTMs. PARP1 ADP-ribosylates lysine residues of the core histone tails. Histone H3 ADP-ribosylation is a PTM competing with H3 methylation by SET7/9. Thus, SET7/9 methylates H1 and H3 in the absence of PARP1. However, the PARylation of H3 stimulates H1 methylation by SET7/9 [61].

The extracellular signal-regulated kinases 1/2 (ERK1/2) promote PARP1 activation through the direct phosphorylation of PARP1 at serine 372 and threonine 373 [62]. The phosphorylation sites are located near the beginning of the BRCT portion of the auto-modification domain, which has been proposed to modulate PARP1 activity [57]. PARP1 and CDK2 specifically interact following hormone treatment via the BRCA1 C-terminal domain (BRCT) of PARP1 and the cyclin-binding domain (CBD) of CDK2, which facilitates the phosphorylation of PARP1 at S785/S786. The phosphorylation of the serines on PARP1 results in an opening of the NAD^+^-binding pocket within the catalytic domain, increasing PARP1 activity, which leads to auto-PARylation. Furthermore, PARP1 and CDK2/cyclin E are recruited to chromatin, resulting in PARylated proteins and the opening of chromatin through the displacement of histone H1. The receptor tyrosine kinase c-Met associates with and phosphorylates PARP1 at Tyr907 (pTyr907 or pY907) [63]. The enzyme activity of c-Met is required for the PARP1 and c-Met interaction, following H_2_O_2_ treatment. The C-Met-dependent PARP1 phosphorylation at Tyr907 increases PARP1 activity and results in a reduced affinity to PARP inhibitors, thereby rendering cancer cells resistant to PARP inhibitors. On the other hand, cancer cells treated with c-Met inhibitors become more sensitive to PARP inhibitors. EGFR is another kinase that phosphorylates PARP1 [63], and the inhibition of both EGFR and PARP1 results in synthetic lethality in highly aggressive triple negative breast cancers (TNBCs). 

## 6. Cancer Proliferation, Metastasis, Angiogenesis: The Role of ADP-Ribosylating Enzymes According to a Deregulated Expression or Use of ART Specific Inhibitors

Various cellular functions are disrupted by the altered expression of ADPR writers, readers, and erasers. Several enzymes involved in ADP-ribosylation are linked to some form of cancer. Specific enzymes may cooperate in one or several forms of intrinsic hallmarks of cancer: uncontrolled proliferation, evasion of growth suppressors, cell death resistance, genome instability, reprogrammed energy metabolism, and escape from replicative senescence [64,65,66,67]. PARPs generating PAR modifications include PARP1/2 and the tankyrases, both of which are targets for cancer therapy. Finally, a paragraph will be focused on MARylating enzymes with specific involvement in certain types of cancer.

### 6.1. PARP1/2/3 Inhibitors in Cancer Treatment

Various cancers were shown to respond well to PARP1/2/3 inhibitors (PARPi). Olaparib, rucaparib, niraparib, veliparib (ABT-888), pamiparib, simmiparib, and talazoparib can be applied for the treatment of breast and ovarian cancers showing *BRCA1/2* mutations, in which lower-fidelity non-homologous end-joining (NHEJ) occurs, resulting in chromosomal aberration and in cell death [64,68]. However, there is continuous research to develop more efficient PARP1 inhibitors. ZINC67913374 is more stable than olaparib when interacting with PARP1 [69]. A novel PARP1 inhibitor, OL-1, was found to induce cell death and to inhibit cell migration in *BRCA1* mutant MDA-MB-436 breast cancer cells [70]. PARP1/2 inhibitors block DNA repair and PARP catalytic activity and trap PARP1 and PARP2 proteins on DNA, interfering with replication and leading to cell death. Tumors displaying molecular features of BRCA-mutant tumors have been indicated with the “BRCAness” phenotype [71]. BRCAness allows for the expansion of the somatic lethality approach to tumors carrying deficiencies in tumor suppressor genes involved in homologous recombination (HR) defective cancers, possessing *BRCA1, BRCA2, ATM, ATR, FANC, RAD51, BRIP1*, or *PALB2* mutations. Cancers with defective ATM protein or aberrations of the MRE11A-RAD50-NBN complex are included in BRCAness. *ATR*, *RAD51, RAD54, DSS1, RPA1, NBN, CHK1, CHK2, FANCD2, FANCA, FANCC, FANCM,* and *BARD1* mutated tumors also respond to PARP inhibitors [72,73].

PARP inhibitors were also tested in *RNASEH2B* deleted cancers, in HER negative cancers, in few cases—as the pretreatment with adjuvant taxanes or anthracycline [73]. Cancers defective in *PTEN*, as prostate cancers, can be sensitive to PARP inhibitors, due to the downregulation of *RAD51.* Since lysine methylase SMYD3, often overexpressed in tumors, regulates BRCA1/2 and DNA repair, the inhibition of SMYD3 with BCI-121 promotes cancer inhibition by PARP inhibitors such as olaparib [70]. Olaparib, as well as similar inhibitors, may induce cancer resistance in patients due to the extrusion by multidrug resistance proteins and ATP-binding cassette pumps. PARP1 inhibitor AZD2281 was more effective when co-administered with the P-glycoprotein inhibitor tariquidar [71,72,73]. The use of PARP inhibitors has been approved for treatment in *BRCA1* and *BRCA2* mutated breast and ovarian cancers, and for therapy in platinum sensitive ovarian cancer patients with defective *BRCA1/2* genes [74]. Olaparib resistance may occur in cancers with homologous recombination defects [73]. Resistance due to the reversion of mutated *BRCA1/2* and other mechanisms is common; these mechanisms include: PARP1 loss of function and the silencing of DNA damage response (DDR) proteins such as 53BP1 and REV7, that reactivate the HR pathway [72,73,74]; selective loss of *PARG* [75,76,77]; loss of function of proteins involved in destabilization of the replication fork (*EZH2, MUS81*) [78]; high expression of CCAAT /enhancer-binding protein β (C/EBPβ), upregulating the HR pathway (BRCA1, RAD51) [79]. Ribonucleotide excision repair requires topoisomerase 1 and PARP1 activities: *RNASEH2B* mutant prostate cancers were found to be hypersensitive to PARP inhibitors [73,80]. In *RAD51C*-silenced high-grade serous ovarian cancers, *RAD51* re-expression was associated with an acquired loss of *RAD51* promoter methylation during PARP inhibitors therapy [81]. The deletion or inhibition of eraser enzymes such as PARG may potentiate synthetic lethality drugs in cancers with mutant phenotypes [42,43,75]. The synthetic lethal interaction between ARH3 and PARG may set the basis for new therapeutic approaches [82]. The loss of *ARH3* was shown as a mechanism to develop PARP1/2 inhibitor resistance [82]. The application of a DNA methyltransferase inhibitor (DNMTi) may re-sensitize tumors to primary therapies [83]. In cells deficient in *PTEN,* overexpressing *SAMHD1* with *53BP1* loss, PARP inhibitors were ineffective [84]. E2F7 is a member of the E2F transcription factor family. E2F7 and E2F8 are atypical E2F family members and mediate transcription repression, particularly of homologous recombination proteins. E2F7 in *BRCA2*-deficient cells confers sensitivity to PARP inhibitors, while its depletion counterbalances the chemosensitivity of *BRCA2*-deficient cells, by promoting both HR and fork stability [85]. 

PARP3 deficiency or inhibition leads to the growth suppression of *BRCA1*-deficient TNBC cells [86,87]. The genetic PARP3 knock-down led to decreased survival and in vivo tumorigenicity, results dependent on a defective Rictor/mTORC2 signaling complex, and to ubiquitination and degradation of Rictor. Accordingly, PARP3 MARylates GSK3β, a positive regulator of Rictor ubiquitylation [86]. PARP3 selective inhibitor ME0328 displays higher selectivity over ARTD1 and its nearest homologs [11]. Non-NAD^+^-like PARP1 inhibitors, binding to sites different from the NAD^+^ binding site, have been studied for prostate cancer treatment [88]. Several conjugates of ADP and morpholino nucleosides target PARP1/2/3 with high specificity [89], while ABT-888 showed high specificity toward PARP2 [90]. A different approach to reverse PARP inhibitors resistance has been proposed through targeting autophagy, and the enzymes controlling autophagy in HR repair proficient breast cancers [91]. Topoisomerase 1 inhibitors have been shown to be effective in cancer therapy of HR-deficient, Schlafen 11-positive cells in association with Olaparib, for the treatment of *BRCA1*-, *BRCA2*-, and *PALB2*-deficient cells [92]. Researchers are continuously developing new scaffolds for the setup of PARP inhibitors [93]. Furthermore, PARP inhibitors’ association with HDAC inhibitors such as SAHA or belinostat have shown benefits in prostate cancer [94]. A list of PARP1/2/3 inhibitors applied to cancer treatment is presented in Table 1.

### 6.2. Tankyrases: Cell Functions and TNKS Inhibitors in Cancer Treatment

High levels of TNKS1 and/or TNKS2 expression have been observed mainly in colon, lung, brain, breast, ovarian, and liver cancers [31]. Several mutations altering PARylation activity may occur, depending on large deletions or point mutations. The stability of many proteins is regulated via PARdU: Axin1/2, PTEN, 3BP2, and TRF1 are processed and degraded by PARdU, as well as the actors of this activity, TNKS1/2 and RNF146 [31]. Tankyrases modify the tumor suppressors phosphatase and tensin homolog (PTEN) and axin 1/2, which regulate the levels of phosphatidylinositol 3,4,5 triphosphate and β-catenin, negatively regulating the Akt/PI3K pathway (PTEN) and Wnt signaling (axin). After PTEN or axin1/2 are PARylated by TNKS1/2 on acidic residues, RNF146 ubiquitylates on lysines, destining them to proteasomal degradation. 

Wnt-driven cancers such as CRC, HCC, and lung cancers may benefit from Tankyrase inhibitors [31,101,102]. In lung cancer, most often showing TNKS overexpression, tumors possess abnormal Wnt activity, due to APC and β-catenin mutations, as well as to the deregulation of upstream Wnt signaling effectors, such as Dishevelled 3 (Dvl-3), or the downregulation of Wnt antagonists, as Wnt-inhibitory factor 1 (WIF 1). Among the substrates of Tankyrases, AXIN is a key regulator of the canonical Wnt signaling pathway. Wnt activation regulates the level and subcellular localization of the β-catenin transcription factor. Glycogen synthase kinase 3β (GSK3β), in collaboration with axin, APC (adenomatous polyposis coli), and other proteins, phosphorylates β-catenin, that is then degraded by the proteasome [102,103,104]. The activated Wnt complex inhibits the AXIN/GSK3β complex and stabilizes β-catenin, which can translocate into the nucleus. β-catenin protein levels are kept low by phosphorylation, so that β-catenin is ubiquitylated by a complex containing a β-transducin repeat-containing protein (βTrCP). Tankyrases PARylate AXIN and recruit the E3 ubiquitin ligase RNF146 containing the PAR-binding WWE domain, which ubiquitylates AXIN, destining it for proteasomal degradation. In this optic, tankyrase promotes Wnt signaling, while TNKS inhibitors are useful in the treatment of Wnt-driven cancers [31,104]. G007-LK is a selective tankyrase inhibitor [105]: the treatment with this inhibitor induced the loss of expression of MYC and impaired cell growth, with the accumulation of β-catenin degradasomes. IWR1 and AZ-6102 are also selective for tankyrase [106]. TNKS1/2 PARylate PTEN. PTEN contains a Tankyrase Binding Motif (TBM) at the N-terminal region, close to the phosphatase domain. RNF146 recognizes PARylated PTEN and performs its ubiquitylation, destining it to degradation by the proteasome. Endometrial cancer presents the highest rate of *TNKS1* and *TNKS2* mutations [31], followed by colorectal cancer, bladder, melanoma, esophagogastric, and prostate cancers. High levels of TNKS1 and/or TNKS2 expression have been found in colon, lung, brain, breast, ovarian, and liver cancers. In the first attempts to develop ART inhibitors, few chemical structures were shown to target PARPs and tankyrases: XAV939 is a promiscuous inhibitor of PARP1 and tankyrases, while PJ34 and UPF1069 are broad PARP/tankyrase inhibitors. In recent years, inhibitors were developed specifically for tankyrases: some of them bind to the nicotinamide subsite (NS) (such as XAV939, RK-582, LZZ-02, AZ1366), or to the adenosine subsite (AS) (IWR-1, OM-1700, OD366, G007-LK and K-756) [107,108,109,110,111,112,113,114,115,116]. Dual binders (DB), recognizing both nicotinamide and adenosine subsites, have been found among PARP1 inhibitors (olaparib): for TNKS, several inhibitors showed a potential therapeutic use (CMP4, WIKI4, TNKS656) [31]. K-756 and RK-287107 block cell growth in COLO-320DM and SW403 cancer cells, carrying a truncated form of APC lacking a short stretch of 20 amino acids involved in β-catenin binding. RKO, DLD-1, and HCC2998 cells, possessing a different, long truncated form of APC, that preserves the binding to β-catenin, do not respond to K-756, but are sensitive to LZZ-02. The proposed approach to treat APC mutated cancers is the use of G007-LK, LZZ-02, or RK-582 at low dosage in association with the PI3K (BKM120) and the epidermal growth factor receptor (EGFR) (erlotinib) inhibitors for colon cancer treatment [31,113]. The proteins targeted by TNKS include vinculins, that anchor F-actin to adherens junctions (AJ), structures involved in cancer development, while TNKS inhibitors prevent the assembling of AJ [117]. 

Tankyrases recruit specific motifs (RxxPDG “hexapeptides”) in their binding partners via an N-terminal region of ankyrin repeats. These ankyrin repeats form five domains termed ankyrin repeat clusters (ARCs), that bind the substrates [118]. Tankyrase partners bind to ARCs by means of RxxPDG hexapeptide motifs and include Disc1, RAD54, Fat4, Striatin, SH3BP1, MERIT40, and BCR [118]. C44 was developed as a TNKS blocker, acting as a protein–protein interaction (PPi) inhibitor. These inhibitors induce the degradation of TNKS1/2. New peptidomimetics that bind to the ankyrin domain of tankyrase have been studied by fragment–based screening, and structures are available [119,120]. A list of tankyrase inhibitors applied to cancer treatment is presented in Table 2.

### 6.3. Inhibitors Specific for MARylating Enzymes with A Role in Cancer Development

#### 6.3.1. Macro-PARPs: ARTD7, ARTD8, ARTD9

ARTD9 (PARP9/BAL1), ARTD7 (PARP15/BAL3), and ARTD8 (PARP14/BAL2) were originally identified as highly expressed in B-aggressive lymphomas (BAL) [121]. The macro-PARPs are characterized by the presence of N-terminal macrodomains; ARTD9 and ARTD7/PARP15 possess two macrodomains, while ARTD8/PARP14 contains three macrodomains. These macrodomains are mono-ADP-ribosylation reader modules. ARTD8/PARP14 MARylates histone deacetylases HDAC2 and HDAC3, as well as TBK1, TANK (TRAF-associated NF-κB activator) binding kinase 1, involved in IRF3 activation and interferon signaling [122]. ARTD8 MARylates STAT1, leading to reduced STAT1 phosphorylation levels: this leads to the suppression, in macrophages, of the IFNγ-STAT1 signaling and of the TNF-α/IL-β proinflammatory pathway [1]. New, potent ARTD8 inhibitors have been developed, such as H10, possessing a more than 20-fold higher selectivity on ARTD8 in respect to PARP1 [123,124,125]. Furthermore, several (Z)-4-(3-carbamoylphenylamino)-4-oxobut-2-enyl amides were developed: among these, compound 4t displays a >10-fold selectivity over ARTD5/PARP5a (TNKS) and a >5-fold selectivity over closely related ARTD10, but is also recognized by PARP1 [123,124,125]. In addition, various diaryl ethers are specific of one of two closely related mARTDs, either for ARTD10 or for ARTD8. Structure-based activity studies showed that compound 8b inhibits ARTD10 at nanomolar concentration, with about 15-fold selectivity over ARTD8. On the other side, compounds 8k and 8m inhibit ARTD8/PARP14 at the nanomolar level, with considerable selectivity over ARTD10 [126]. The ARTD8 macrodomain 2 inhibitor may be applied in cancer therapy. While there is no inhibitor specific for a specific macrodomain, the compound GeA-69, interacting with the ARTD8 macrodomain 2, has been studied in lymphoma and myeloma, as well as for the treatment of asthma [127]. Ribon Therapeutics are studying the effects of their ARTD8 inhibitors: in vitro data showed that ARTD8 plays an immune-suppressive role in the tumor microenvironment, suggesting that ARTD8 targeting could generate an anti-cancer inflammatory response, similarly to results obtained by means of checkpoint inhibition [123,124,125,126,127,128,129]. A list of Macro-PARP inhibitors and application prospects to cancer treatment is presented in Table 3.

#### 6.3.2. Other MARylating ART Enzymes Linked to Cancer and Potential Enzyme Inhibitors

The relationship between ART enzymes and cancer has been observed through the finding of overexpressed genes in cell lines or tissues and by the use of ART inhibitors. In the first years, the developed inhibitors were designed targeting a group of ART enzymes; in recent years, novel compounds have been developed specifically for a single ART enzyme, with effects in the nanomolar concentration. For instance, there are new inhibitors targeting just ARTD10 [126,130,131,132,133], ARTD11 [128,134,135,136,137,138,139,140,141,142,143,144,145,146,147], or ARTD14/PARP7 [145,146,147].

ARTD10 has tumor suppressor activity by inhibiting myc and by the MARylation of Aurora A kinase, inhibiting its phosphorylation on the same residue: The MARylation is exerted after interacting with the ubiquitin ligase RNF114, which ubiquitylates ARTD10, activating it; a loss of RNF114 in cancers (mutation, deletion, downregulation) may lead to a loss in ARTD10 activity [1,133]. ARTD10 may also have oncogenic activity, being involved in some cancers, promoting the proliferation and restart of stalled replication forks, and alleviating cell survival during replication stress [1,148].

However, to complete a MARylation cycle, the writers and the eraser should be expressed at similar levels. MAR hydrolases cleave the modifications promoted by ARTD10. MacroD2, expressed in neuroblastomas, and MacroD1, overexpressed in various cancers, can counteract the tumor suppressive role of ARTD10 [1,49]. The MARylation of ARTD10 is recognized by ARTD8/PARP14, that is docked to the MAR structure to form a protein complex. Several molecules have shown the potential to selectively inhibit ARTD10 [14,130,131,132,133], such as the cell-permeable OU135, a 3,4-dihydroisoquinolin-1(2H)-one that contains a methyl group at the C-5 position and a substituted pyridine at the C-6 position [131], and 4-benzyloxybenzimide derivatives [130]. The targets of ARTD10 that could require a fine-tuning of MARylation are glycogen synthase kinase GSK3β, NEMO/IKK-γ, a subunit of NF-kB transcription factor, histone H3, and the polo-like kinase PLK1 [1]. Targeting ARTD10 could decrease the proliferation of certain types of cancer and HCC cells overexpressing ARTD10.

ARTD11 MARylates ubiquitin E3 ligase β-TrCP, regulating the IFN-I-dependent antiviral activity [135]. ARTD11 inhibitors showed the release of ARTD11 from nuclear envelope complexes, where it associates with NUP153 [136] (Figure 3). In addition, ARTD11 was shown to interact with ARTD12 and to inactivate the replication of the Zika virus through the degradation of NS1 and NS3 viral proteins [137].

ARTD14/PARP7 is amplified and highly expressed in squamous cell carcinomas, in ovarian and lung tumors, and associated with poor survival [138,139,140]. The MARylation of Protein substrates such as α-tubulin, by ARTD14/PARP7, has a role in microtubule control in ovarian cancer [141,146,147]. The enzymes are also involved in the maintenance of stemness and pluripotency [142]. PARP-7 MARylates immune-relevant protein targets and modulates cancer-directed host immune responses [143]. RBN-2397’s specific inhibition of ARTD14 suppresses cell proliferation and activates the type-I IFN pathway in cancer cells [144,145,146,147]. RBN-2397 is in a phase 1 clinical trial for patients with advanced solid tumors (Identifier: NCT04053673). The ARTD14/PARP7 dependency for proliferation of cancer cells (i.e., lung cancer) and cells with high baseline expression of interferon (IFN)-stimulated genes. Interestingly, RBN-2397 enhanced IFN signaling and induced both cancer-cell-autonomous and immune-stimulatory effects [145,146,147].

ARTD15/PARP16 associates with the endoplasmic reticulum (ER) [149,150,151,152]. It is a single-pass transmembrane protein with the N-terminal region (amino acids 1–280) facing the cytoplasm and a C-terminal tail facing the ER lumen. ARTD15 interacts with the nuclear transport factor karyopherin-1/importin-1 (Kap-1β), MARylating it [150]. ARTD15 auto-modifies itself and MARylates the double-stranded RNA-dependent protein kinase (PKR)-like ER kinase (PERK) and the inositol-requiring enzyme 1α (IRE1α) [150,151]. By this modification, ARTD15 activates PERK and IRE1α, two proteins relevant for the ER stress response and for the unfolded protein response (UPR). During ER stress, ADP-ribosylated PERK and IRE1α increase their kinase activities as well as IRE1α endoribonuclease activity. ARTD15 inhibitors based on latonduine analogs have been validated and tested to correct mutated CFTR by blocking IRE1α modification [153,154]. ARTD15 is associated with UPR-dependent inflammation, involved in inflammatory diseases [6]. Through Kapß1 MARylation, ARTD15 regulates and controls nucleo-cytoplasmic trafficking. Inhibitors targeting the site of ADP ribosylation on Kapß1 and the ARTD15 catalytic site are potential drugs for innovative therapeutic strategies.

ARTD17/PARP6 is an enzyme involved in some cancers. It has been identified as a negative regulator of cell-cycle progression in HeLa cells: ARTD17 overexpression was reported to arrest cells in the S-phase; this activity is dependent on a functional catalytic domain. In colorectal cancer, ARTD17 promotes cancer growth [136]. In addition, ARTD17 exerts proliferative effects, while its inhibition induces the formation of multipolar spindles in breast cancer [155,156,157,158]. By screening a library of compounds for the ability to induce mitotic defects, researchers identified AZ0108 as a potent ARTD17 inhibitor, showing antitumor effects in vivo and inducing cell death in breast cancer cells in vitro [144,157].

Concerning cholera-toxin-like ARTs, ARTC1 MARylates the Grp78/BiP chaperone, required for protein quality control in endoplasmic reticulum [159]; this modification is relevant for the ER stress response system. Either alterations in UPR or in ER stress response can have consequences in normal cells and in cancer development. ARTC1 regulates the RhoA/ROCK/AKT/β-catenin pathway in colon carcinoma, possibly through the regulation of kinase signaling pathways by the direct ADP-ribosylation of key kinases [160,161,162,163].

A list of MARylating enzymes and their potential inhibitors applied to cancer treatment is presented in Table 4.

### 6.4. Sirtuin Inhibitors Addressing the Roles of MARylating Sirtuins in Cancer

SIRT1, SIRT2, SIRT3, and SIRT6 are blocked by a sirtuin inhibitor selisistat (Ex527) targeting the NAD^+^ binding pocket; the sirtuins are not inhibited by PARP inhibitors such as olaparib and similar compounds, PJ34, and XAV939, due to differences in their NAD^+^ binding site [166,167].

SIRT4 is localized in the outer mitochondrial membrane. SIRT4 MARylates GDH [17] and deacylates pyruvate dehydrogenase (PDH), two enzymes that regulate the tricarboxylic acid (TCA) cycle. The MARylation of GDH inhibits the conversion of glutamate to α-ketoglutarate (α-KG), decreasing the glutamine uptake: α-KG upregulates H3/H4 histone acetylases and stimulates histone lysine demethylases; on the other hand, during the TCA cycle, the accumulation of fumarate and succinate leads to the inhibition of α-KG-dependent enzymes, while an increase in citrate inhibits α-KG-dependent N6-methyladenine demethylase (ALKBH5). SIRT4 was also associated with negative impacts on the mitochondrial quality and with aging [27].

SIRT6 was found to be overexpressed in skin cancer and non-small cell lung cancer (NSCLC) [168,169], osteosarcoma, colon carcinoma, serous ovarian cancer, and clear cell renal cell carcinomas (ccRCC) [169,170,171,172,173], showing a poor prognostic value. However, in certain types of cancer, SIRT6 is classified as a tumor suppressor. Since SIRT6 deacetylase activity has important effects in cells and in cancers, and since NAD^+^-binding inhibitors block both deacetylation and MARylation reactions, it is not possible to discriminate between the two activities in SIRT6 studies [174]. SIRT6-silenced melanoma cells showed a considerable anti-proliferative effect both in vitro and in vivo [175]. SIRT6 has been involved in genome integrity, DNA repair, energy metabolism, and inflammation. SIRT6 levels decrease during aging and cell senescence [176].

SIRT6 MARylates itself, as well as PARP1 [177,178,179,180,181,182,183,184,185,186], enhancing DNA repair, especially when phosphorylated by JNK on Ser10. SIRT6 ADP-ribosylates lysine demethylase JHDM1A/KDM2A, with a role in the epigenetic modification of histones [181]; SIRT6 MARylates the chromatin silencing factor KAP1 (nuclear co-repressor protein KRAB associated protein 1), fine-tuning the interaction of KAP1 with the heterochromatin protein HP1α: this leads to the silencing of LINE1 retrotransposons [179]. SIRT6 MARylates BAF170, which activates the transcription of Nrf2 target genes: this activation regulates the boost of the mitochondrial function dependent on Nrf2 [182].

SIRT6 functions upon DNA damage, and activates two DNA repair pathways, the nonhomologous end joining (NHEJ) and the base excision repair (BER). SIRT6 induces, by a protein–protein interaction, the formation of a multiprotein complex (SIRT6, RPA, Ku70, Ku80, DNA-PKcs, ISWI/SNF2H) [187,188,189,190]: the complex recruits BRCA1, 53BP1, CDH4, and PARP1. SIRT6’s MARylating activity induces the induction of the p53- and p73-dependent apoptotic pathways in cancer cells [178]. SIRT6 associates in a phosphorylation-dependent form with the Ras-GTPase activating protein 1 (G3BP1) [177], with transcription factors BCLAF1, NKRF, and THRAP3, the telomerase regulator YLPM1, and the RNA polymerase complex factors COIL and XRN2. The development of small molecules inhibiting specifically either deacetylation or MARylation may provide new clues on SIRT6 functions [168,183,184].

A study on the structure of bacterial SIRT bound to the acetylated +2 arginine peptide showed that the arginine entered in the active site and reacted through a deacetylation reaction intermediate, yielding an ADP-ribosylated peptide [167]. New compounds, such as diketopiperazine-containing 2-anilinobenzamides, simultaneously targeting the “selectivity pocket” substrate-binding site and the NAD^+^-binding site [180], effective in SIRT2 inhibition, have been set up as the basis to develop new specific inhibitors [191,192].

SIRT7 has been involved in genome integrity and non-homologous end-joining (NHEJ) DNA repair [96]. SIRT7 has auto-MARylation activity [39]. SIRT7’s auto-MARylation occurs on several sites: proteomic studies identified 7–8 MARylated peptides. Auto-MARylation modifies SIRT7’s chromatin distribution. The ELHGN catalytic motif that is conserved among sirtuins, H187, is involved in deacetylation and faces the NAD^+^-binding pocket and the catalytic site, together with the flanking residues E185 and N189. In SIRT6 and SIRT7, E185 and N189 are in the opposite direction, facing the surface of the cavity, and the two residues interact forming a loop. These residues are important for their role in the ADP-ribosylation reaction: E185 is the catalytic residue that starts the reaction, while N189 is the first acceptor of the ADP-ribosyl moiety [39]. In Table 5 a list of candidate sirtuin inhibitors is presented.

## 7. Discussion and Perspectives

In this review the most studied players of MARylation and PARylation have been reviewed. The main objective was to relate the proteins targeted by these post-translational modifications and their interactions with specific proteins toward a disruption of their roles in the maintenance of cell functions through MARylation/PARylation inhibitors. These disfunctions lead to cell proliferation, anomalies in growth control, loss of cell-to-cell contacts, and tumor development. Furthermore, worthy data have been produced through clinical trials on several types of PARP1/2/3 inhibitors, on tankyrase/PARP5a/b inhibitors, and on MARylation inhibitors targeting ARTD14/PARP7. The deregulated cell functions may derive from the MAR/PAR cycling control, stemming from the overexpression of eraser enzymes that have been associated with various cancers. Approaches using PARP1/2 inhibitors or broad PARP inhibitors have been proposed and tested not only in cancer treatment [98,99,100], but also for non-oncological diseases: for instance, in the protection against oxidative stress, in the suppression of inflammatory responses, in the treatment of ALS and neurodegenerative diseases, and in TDP-43 dependent neurological disorders [194,195,196]. Thus, PARP inhibitors are seeing a new wave of medical exploitation both in cancer and in inflammatory diseases, since it can be applied in the therapy of non-oncological diseases.

Similarly to MAR erasers, several RNA viruses express nsp3, a macrodomain protein with eraser activity that subverts the stress granule structure and re-establishes the translation and function of stalled mRNAs [197].

Recently, new data have been produced on MacroD1, MacroD2, and TARG, uncovering their interactomes and their cell specificity [198,199]. Among these findings, the specific cellular localization of the three eraser proteins, the mitochondria for MacroD1, the nuclei, nucleoli and stress granules for TARG1, and the nuclei and cytoplasm for MacroD2, were found to be expressed mainly in tissues of neural origin.

Furthermore, compounds able to inhibit and discriminate between ARH3 and ARH1 have been developed and could be tested to protect cells from the deregulated MAR/PAR hydrolytic activity or the inhibition of the nsp3 macrodomain of SARS-CoV2 [199,200].

The perspectives for ADP-ribosylation’s future studies lead in the direction of antiviral therapies and the development of more specific inhibitors of the MAR/PAR modifications and their control, for the amelioration of oncological and non-oncological diseases.

Finally, a new enzyme with supposed PARylating activity has been associated with mitochondria: Neuralized-like protein 4 (NEURL4) possesses an H-Y-E-like ART domain in the C-terminal domain, in addition to six Neuralized Homology Repeat (NHR) domains, functioning as protein–protein interaction scaffolds [201,202,203]. The PARylating activity associated with mitochondria is lost in cells lacking NEURL4. One main protein target for PARylation is mitochondrial Ligase III (mtLIG3), required for mtDNA stability. Furthermore, LRRC9-ART has been characterized at the expression level, and is predicted to interact with ZFP36L2, an RNA-binding protein that controls the cell cycle and is involved in pancreatic cancer [201,202]. New macrodomain proteins have also been identified, such as C12ORF4, a cytoplasmic protein predicted to have eraser activity, involved in mast cell degranulation [203]. C12ORF4 is highly expressed in glioblastomas, adenocarcinomas of the ovary and pancreas, and lymphomas. For a more rapid circulation of the new findings within the PARP/ART community, it is important to update the databases on the enzymes involved in this PTM, such as ADPriboDB 2.0 [204]. Just as importantly, also in the plant kingdom, a sirtuin enzyme has been reported to possess MARylating activity [205]. New findings will support MAR/PAR studies for a deeper understanding of their functions.

Further studies analyzing 3D structures of MARylation/PARylation enzymes should help develop new inhibitors. It is envisaged that in the next year new therapies will be tested via the exploitation of PARylation and MARylation inhibition, the safety of the administration of new drugs, and the recovery to a healthy state.

## Figures and Tables

**Figure 1 ijms-22-10829-f001:**
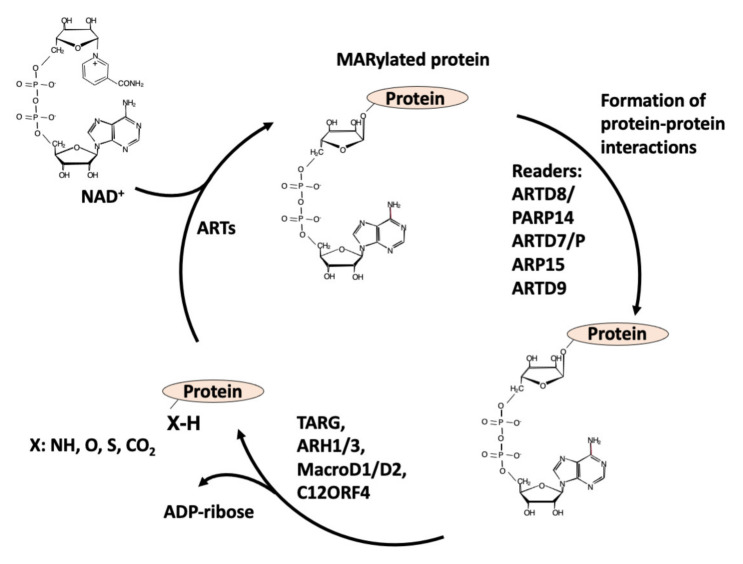
Writers, readers, and erasers of MARylation of proteins.

**Figure 2 ijms-22-10829-f002:**
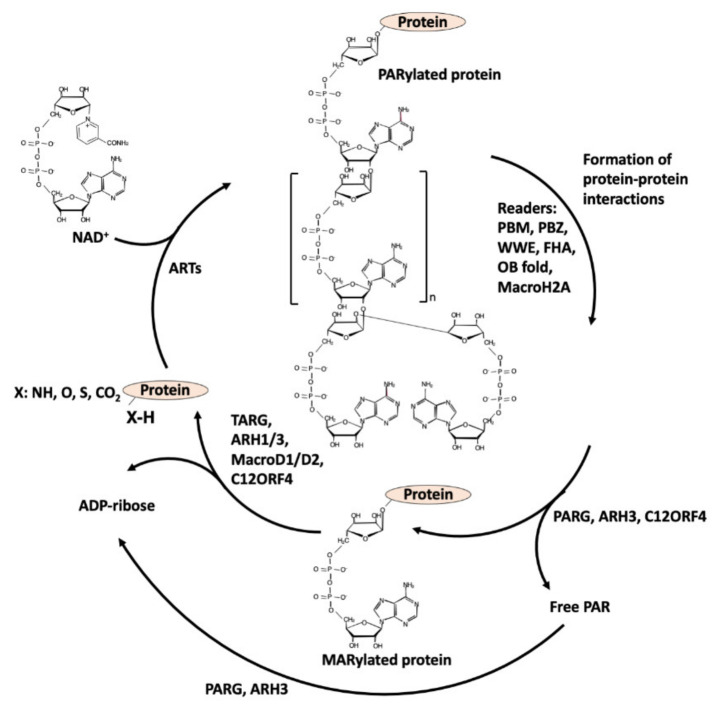
PARylation cycle: writers, readers, and erasers.

**Figure 3 ijms-22-10829-f003:**
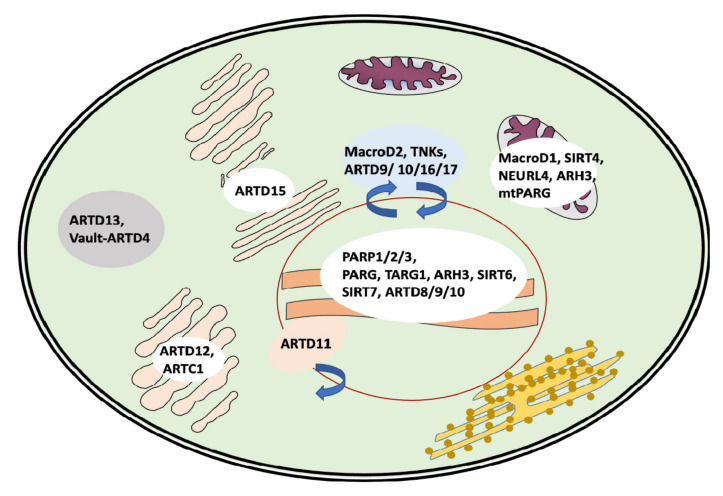
Intracellular localization and shuttling of the major players of the MARylation/PARylation cycles.

**Table 1 ijms-22-10829-t001:** Selective inhibitors of PARPs (PARP1/2/3) and therapeutic potential.

Drug	Target Molecules	Therapeutic Applications	References
Olaparib, rucaparib, niraparib, veliparib, talazoparib, often associated with alkylating agents or platinum	PARP1/PARP2 inhibition and trapping on DNA	*BRCA1, BRCA2, ATM, ATR, FANC, PALB2* mutated, *CCDC6* inactivated, HR defective cancers; *RNASEH2B* deleted cancers; HER negative cancers, prostate cancer, 53BP1 deficient cancers	[14,67]
Non-NAD^+^-like PARP-1 inhibitor, 5F02	PARP1	Prostate cancers (hormone dependent and independent); PARG mutated cancers	[88]
Veliparib + SAHA or belinostat	PARP1 and HDAC	Prostate cancer	[94]
Guadecitabine + talazoparib	PARP1 and DNMT1	Breast and ovarian cancers	[83]
SMYD2 inhibitor LLY-507 or BCI-121 + olaparib	PARP1 and SMYD2	Serous ovarian cancer, high grade (HGSOC)	[57,58]
PARPi + EZH2 inhibitors	*BRCA1*-mutant cell sensitization to PARPi	Inhibition of PARP by PARPi attenuates alkylating DNA damage-induced *EZH2* downregulation, promoting EZH2-mediated gene silencing and cancer stem cell control	[78]
PARPi as chemosensitizer in combination with temozolomide, DNA crosslinkers (cisplatin), or bleomycin	PARP1/2	BRCAness in various cancers	[71,73]
PJ34 ((N-(6-oxo-5, 6-dihydrophenanthridin-2-yl)-N, N-dimethylacetamide), veliparib	Broad inhibitors	Protection against oxidative stress; suppression of inflammatory responses, ALS, TDP-43 degenerative processes	[95,96,97]
ME0328	ARTD3/PARP3	Exploits vulnerabilities in DNA repair in cancers	[87]
5-Amino-7-(aldehyde)-2[(naphthalene-2-yloxy)methyl]-[1,3,4]thiadiazolo-[3,2-α]-pyrimidine-6-carbonitrile scaffolds	PARP1		[11,93]
Conjugates of ADP and morpholino nucleosides	PARP1/2/3	Cancers	[89]
ABT-888	PARP2	Epithelial ovarian cancer	[90]
Azaquinolones	PARP1	Cancers	[98]
Veliparib + busulfan	PARP1	Myeloproliferative neoplasms	[99]
Olaparib + radiotherapy	PARP1	Medulloblastoma	[100]

**Table 2 ijms-22-10829-t002:** Tankyrase inhibitors applied to cancer treatment: inhibitors stabilize and increase Axin levels and decrease nuclear β-catenin levels.

Drug	Target	Therapeutic Applications	References
IWR-1, AZ-6102	ARTD5, ARTD6 (TNKS)	Wnt-driven cancers Colon, lung, brain, breast, ovarian, liver cancers.	[106,107]
N-([1,2,4]triazolo[4 ,3-a]pyridin-3-yl)-1-(2-cyanophenyl)piperidine-4-carboxamide (TI-12403)	TNKS	Combined 5-FU/TI-12403 treatment synergistically block proliferation of colorectal cancer (CRC)	[108]
M2912	TNKS	CRC	[109]
Spiroindoline derivative 40c RK-287107)	TNKS	CRC	[110]
MSC2504877 + CDC4/6 inhibitors	TNKS	*APC* mutant CRC	[111]
NAM binders			
XAV939, LZZ-02, AZ1366, RK-582, RK-287107	TNKS	CRC with *APC* mutations, Lung cancers	[112]
In association with PI3K (BKM120) and epidermal growth factor receptor (EGFR) (erlotinib) inhibitors		CRC with *APC* mutations	[113]
Adenosine binders			
IWR-1, G007-LK, OD366, OM-1700, K-756	TNKS: restoration of β-catenin degradation	CRC	[103]
Dual binders: binding to adenosine and nicotinamide subsites			
CMP4, WIKI4, IWR-1, IWR-2, CMP4b puinazolinone, CMP24, [1,2,4]triazol-3-ylsulfanylmethyl)-3-phenyl-[1,2,4]oxadiazoles, NVP-TNKS656	TNKS	CRC, mammary adenocarcinoma	[106,114]
EB-47	TNKS, PARP1		[114]
Quinoxaline-based set of fragments Ankyrin repeat cluster (ARC)-TNKS binding modules (TBM)	TNKS		[118,119]
C44 disrupts the protein–protein interaction (PPI) interface of TNKS and USP25	TNKS	Prostate cancer, increase of Axin-1 levels	[120]

**Table 3 ijms-22-10829-t003:** Macrodomain containing ARTDs: ARTD8 inhibitors for cancer treatment.

Drug	Target	Therapeutic Applications	References
H10	ARTD8/PARP14	In vitro	[123]
4t and 8b, in a group of -4-(3-carbamoylphenylamino) 4 oxobut-2-enyl amides	ARTD8/PARP14	Multiple myeloma, HCC, prostate cancer	[123,124,125]
N-(2(-9H-carbazol-1-yl)phenyl)acetamide/sulfonamide, GeA-69	ARTD8/PARP14 macrodomain	Lymphoma (DLBCL), HCC	[125,127]
8k and 8m diaryl ethers	ARTD8/PARP14	In vitro	[123,124,125]
RBN012759, RBN011980, RBN012759, RBN012811, RBN012042, RBN010860, AZ12629495, RBN0120420; RBN013527 similar to RBN012811, but does not cause the degradation of PARP14	ARTD8/PARP14	In vitro	[128,129]

**Table 4 ijms-22-10829-t004:** Other MARylating ART enzymes: potential inhibitors applied to cancer treatment.

Drug	Target	Therapeutic Applications	References
4-Benzyloxybenzimide derivatives, 4-4-cyanophenoxybenzamide, 3-4-carbamoylphenoxybenzamide, OUl35	ARTD10	Neurodegenerative disorder’ gene amplification in breast and ovarian cancer	[14,95,126,127,128,129,130,131,132,164,165]
ITK7	ARTD11	Causes PARP11 to dissociate from the nuclear envelope	[128,134,143]
RBN-2397 RBN011147 (FRET), RBN11198 (BRET), RBN010860	ARTD14/ PARP7	Metastatic solid tumors, lung cancer, anti-proliferative, macrophage polarization, immune signaling	[133,134,135,136,137,138]
EGCG RBN010860, RBN012148, talazoparib	ARTD15/ PARP16	ER stress, intracellular trafficking, SCLC	[6,66,97,130,139,140]
RBN010860, AZ0108	ARTD17/ PARP6	CRC, various cancers	[138,139,140,141,142,143,144]
Cholera-like ARTs			
Aspecific MARylation inhibitors FRET (RBN011147) and NanoBRET (RBN011198) probes	ARTC1	Various cancers (CRC, glioma), angiogenesis, block of epithelial-to-mesenchymal transition (EMT)	[15,66,118,144,159,160,161,162,163]

**Table 5 ijms-22-10829-t005:** MARylating sirtuins: inhibitors and activators.

Drug	Target	Therapeutic Applications	References
Acetylated lysine-ADP-ribose conjugates	SIRT7	NSCLC, control of ERK1 activity	[183]
EX527 (selisistat), OSS_128167	SIRT6 (and other SIRTs) target the NAD^+^ binding pocket	Chemosensitizing effects	[166,174]
Quinazolinedione derivatives 1-phenylpiperazine derivative	SIRT6, target the NAD^+^ binding pocket	Increase in DNA damage/cell death induced by olaparib in Capan-1 cells, silencing blocks proliferation and induces apoptosis in melanoma	[168,175]
Quinazolinediones, salicylate-like structure OSS_128167	SIRT6 inhibition	Sensitivity of cancer to gemcitabine, pancreatic ductal adenocarcinoma	[191,192]
MDL-800, CL5D, MDL-801	SIRT6 activators	Several cancer cells, antiproliferative	[172,189]
FK866 (NAMPT inhibitor) NAD salvage pathway block	SIRT4 inhibition by NAM	Improving aging and neurological diseases, decreasing oxidative stress	[1,27]
Nicotinamide riboside, miR-424-5p antagomiRs	SIRT4 activation/upregulation	Esophageal squamous cell carcinoma (ESCC) proliferation inhibition	[1,193]

## Data Availability

Data can be found in ADPriboDB 2.0.

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
