# Peer review of "ADP-Ribosylation as Post-Translational Modification of Proteins: Use of Inhibitors in Cancer Control"

_ijms, 2021, doi:10.3390/ijms221910829_

Round 1

Reviewer 1 Report

The review presented here nicely summarizes the current situation of the various mechanisms associated with ADP-ribosylation and, with its many sophisticated tables, presents the wide range of existing inhibitors of these enzymes. 

However, I would like to highlight some minor revision points.
First, the abbreviations in the text should be adjusted. The term ADP-ribose is used in the text only as an abbreviation. I therefore ask you to check the abbreviations again.
Besides the inhibitory function of ADP-ribosylation by "readers", steric inhibition is also a possibility. I ask to discuss this briefly.
Another point is the description of the mitochondrial sirtuins. SIRT4 is known to inhibit GDH as well as PDH (line 98). Likewise, no inhibitors of mitochondrial sirtuins are reported. I would see this as only a marginal point, however, these proteins are explicitly mentioned in several figures (e.g. Fig. 3). If they appear in the figures but are clearly underrepresented in the text, this is unpleasant to read.

Again, these are just minor revision points and I think this is a clearly written and nice review. Thank you very much for this excellent work.

Author Response

Dear reviewer, we appreciated your suggestions and revised accordingly.  Some abbreviation has been revised, while introduction already reports  “ to transfer the ADP-ribose (ADPR)”

 Besides the inhibitory function of ADP-ribosylation by "readers", steric  inhibition is also a possibility. I ask to discuss this briefly.

Author: Thank you. We added a sentence: Since there are several motifs recognising specific structures, some of them may compete with other PAR binding proteins, inhibiting their interactions; steric inhibition is another mechanism that may occur to impede the PAR binding, making inaccessible the region to the interacting readers.

Another point is the description of the mitochondrial sirtuins. SIRT4 is  known to inhibit GDH as well as PDH (line 98). Likewise, no inhibitors  of mitochondrial sirtuins are reported. I would see this as only a  marginal point, however, these proteins are explicitly mentioned in  several figures (e.g. Fig. 3). If they appear in the figures but are  clearly underrepresented in the text, this is unpleasant to read.

Author: Thank you: we added a line in table 5 for SIRT4, and this sentence

SIRT4 is localized to the outer mitochondrial membrane. SIRT4 MARylates GDH [17] and deacylates pyruvate dehydrogenase (PDH), two enzymes that regulate the tricarboxylic acid (TCA) cycle. The MARylation of GDH inhibits the conversion of glutamate to α‐ketoglutarate (α‐KG), decreasing glutamine uptake: α‐KG upregulates H3/H4 histone acetylases, and stimulates histone lysine demethylases; on the other side, during the TCA cycle, accumulation of fumarate and succinate leads to inhibition of α‐KG dependent enzymes, while an increase in citrate inhibits α‐KG‐dependent N6‐methyladenine demethylase (ALKBH5). SIRT4 was also associated with negative impacts on the mitochondrial quality and with aging [27].

Reviewer 2 Report

General comments:

This review presents an update on the function of a large set of enzyme “writers”, the “readers” that are recruited by the modified targets, and the “erasers” that reverse the modification to the original amino acid residue, removing the covalent bonds formed. They all provided a comprehensive Tables for providing information of inhibitors in regulating PARPs.

Major comments:

In the end, it will be better to provide the perspective of this direction for ADP-ribosylation future studies.

Minor comments:

  1. Abstract: Please provide the full names with their abbreviations for PTM, MAR, PAR.
  2. Table 5: Sirt changes to SIRT

Author Response

 In the end, it will be better to provide the perspective of this  direction for ADP-ribosylation future studies.

Author: Thank you for your positive feedback. We added a sentence

The perspectives for ADP-ribosylation future studies lead in the direction of antiviral therapies, in the development of more specific inhibitors of the MAR/PAR modifications and their control, for the amelioration of oncological and non-oncological diseases.

Minor comments:

Abstract: Please provide the full names with their abbreviations for  PTM, MAR, PAR.

Table 5: Sirt changes to SIRT

Done, corrected.

Abstract:

Among post-translational modifications of proteins, ADP-ribosylation has been studied for over fifty years, assigning to this post-translational modification (PTM) a large set of functions, including DNA repair, transcription and cell signaling. This review presents an update on the function of a large set of enzyme writers, the readers that are recruited by the modified targets, and the erasers that reverse the modification to the original amino acid residue, removing the covalent bonds formed. In particular, the review provides details on the involvement of the enzymes performing monoADP-ribosylation /polyADP-ribosylation (MAR/PAR) cycling in cancers
